# Lysophosphatidic Acid Accelerates Bovine In Vitro-Produced Blastocyst Formation through the Hippo/YAP Pathway

**DOI:** 10.3390/ijms22115915

**Published:** 2021-05-31

**Authors:** Bo Yu, Helena T. A. van Tol, Christine H. Y. Oei, Tom A. E. Stout, Bernard A. J. Roelen

**Affiliations:** 1Farm Animal Health, Department Population Health Sciences, Faculty of Veterinary Medicine, Utrecht University, 3584 CM Utrecht, The Netherlands; b.yu@uu.nl (B.Y.); h.t.a.vanTol@uu.nl (H.T.A.v.T.); C.H.Y.Oei@uu.nl (C.H.Y.O.); 2Equine Sciences, Department Clinical Sciences, Faculty of Veterinary Medicine, Utrecht University, 3584 CM Utrecht, The Netherlands; t.a.e.stout@uu.nl; 3Embryology, Anatomy and Physiology, Department Clinical Sciences, Faculty of Veterinary Medicine, Utrecht University, 3584 CL Utrecht, The Netherlands

**Keywords:** bovine, lineage segregation, LPA, YAP, CDX2, trophectoderm

## Abstract

The segregation of trophectoderm (TE) and inner cell mass in early embryos is driven primarily by the transcription factor CDX2. The signals that trigger CDX2 activation are, however, less clear. In mouse embryos, the Hippo-YAP signaling pathway is important for the activation of CDX2 expression; it is less clear whether this relationship is conserved in other mammals. Lysophosphatidic acid (LPA) has been reported to increase YAP levels by inhibiting its degradation. In this study, we cultured bovine embryos in the presence of LPA and examined changes in gene and protein expression. LPA was found to accelerate the onset of blastocyst formation on days 5 and 6, without changing the TE/inner cell mass ratio. We further observed that the expression of *TAZ* and *TEAD4* was up-regulated, and YAP was overexpressed, in LPA-treated day 6 embryos. However, LPA-induced up-regulation of CDX2 expression was only evident in day 8 embryos. Overall, our data suggest that the Hippo signaling pathway is involved in the initiation of bovine blastocyst formation, but does not affect the cell lineage constitution of blastocysts.

## 1. Introduction

During mammalian development, the first cell lineage specification is initiated at the compact morula stage, resulting in the formation of a blastocyst with an outer trophectoderm (TE) and an inner cell mass (ICM). During lineage segregation, the transcription factor CDX2 is crucial for proper TE specification [1], whereas OCT4, NANOG, and SOX2 are core regulators of ICM formation [2]. However, the mechanisms that control the expression of these transcription factors are not well understood.

The Hippo signaling pathway is highly conserved in mammals and was initially described as a pathway involved in specifying organ size by controlling cell proliferation and apoptosis [3,4,5,6]. Recently, studies have revealed that the Hippo pathway is involved in follicular activation and follicle growth [7,8,9]. In addition, emerging evidence has indicated that the Hippo-YAP signaling pathway also plays an essential role in controlling the expression of the key transcription factors in first-lineage segregation (ICM and TE differentiation) during mouse preimplantation development [10,11,12,13]. In the cells on the outside of the morula, where the Hippo pathway is inactive, YAP/TAZ remain unphosphorylated and migrate to the nucleus to bind the transcriptional coactivator TEAD4 facilitating *Cdx2* expression, and subsequently driving them to a TE fate [11,13,14]. Moreover, it has been reported that both maternal and zygotic-derived YAP promote CDX2 expression, and that *Tead4*-deleted mouse embryos fail to develop TE [10,15]. When the Hippo pathway is active, YAP/TAZ are phosphorylated, remain cytoplasmic and become ubiquitinated. As a result, there is no *Cdx2* expression in these cells, driving them towards an ICM fate [16,17].

Interestingly, unlike in the mouse, YAP is localized to the nucleus of both the ICM and TE cells of human blastocysts. In addition, YAP overexpression has been reported to promote naïve pluripotency in human embryonic, and induced pluripotent, stem cells [18]. These data indicate that role of the Hippo/YAP pathway in first-lineage determination differs among mammalian species.

Lysophosphatidic acid (LPA) is a bioactive phospholipid that regulates a broad range of cellular effects by activating specific G protein-coupled receptors [19,20,21]. It has been demonstrated that LPA-mediated signaling plays a crucial role in embryo spacing and the timing of implantation in mice [22,23]. Recent studies have demonstrated that LPA inhibits the Hippo pathway kinases LATS1 and LATS2 via Ga12/13-coupled receptors [18,24,25]. Phosphorylation of YAP/TAZ via LATS1/2 leads to ubiquitination-dependent degradation [24,26], a process that is inhibited by LPA.

To understand the role of the Hippo signaling pathway in bovine preimplantation embryo development, we cultured bovine in vitro produced embryos in the presence of 10nM LPA from the timing of fertilization. The results suggest that in the presence of LPA, blastocyst formation starts earlier, YAP expression is up-regulated and CDX2 expression is subsequently increased. Combined, the data suggest that in bovine embryos the Hippo/YAP signaling pathway plays an important role in blastocoel formation and the first cell lineage segregation event.

## 2. Results

### 2.1. LPA Accelerates Blastocyst Formation

To examine how the Hippo pathway is involved in embryo development, bovine oocytes were fertilized in vitro and then cultured to the day 8 blastocyst stage in the presence of 10^−5^ M LPA. Blastocyst formation was first detected on day 6 in the control group. In the presence of LPA, however, blastocysts were already formed at day 5 (Figure 1A). In addition, the blastocyst percentage on day 6 was 14.1% in the control group, whereas in the LPA group it had increased to 26.6% (*p* < 0.01). A consistently higher (*p* < 0.05) expanded blastocyst percentage was detected on day 6 for embryos cultured with LPA (5.4%), compared to control embryos (1.3%) (Figure 1B). However, the percentages of both blastocysts and expanded blastocysts on days 7 and 8 of culture were similar between embryos cultured with and without LPA (Figure 1A,B). Similar to our observation on days 7 and 8 of embryo culture, no significant differences were found in the percentages of uncleaved zygotes, 2–8-cell embryos and >8-cell embryos at day 5 of culture (Figure 1C).

To further determine whether LPA stimulation only affects the developmental status of embryos on days 5 and 6 and is time-dependent, embryos were first cultured until day 5 in the absence of LPA; >8-cell stage embryos were then randomly selected and cultured in the presence of LPA. The blastocyst percentage from these >8-cell embryos was not significantly different between the control group (42.1%) and embryos stimulated with LPA for 2 h (37.0%) or 4 h (43.9%), whereas the blastocyst percentage was significantly higher (*p* < 0.05) when the embryos were exposed to LPA for 24 h (56.8%) (Figure 1D). These data indicate that LPA accelerates blastocyst formation at around day 5 and may therefore affect the first lineage segregation event.

### 2.2. Gene Expression Levels of LPA-Cultured Embryos

To evaluate how LPA accelerates blastocyst formation, we next quantified gene expression for *YAP*, *TAZ*, *TEAD4* and of the lineage-specific genes *CDX2*, *OCT4* and *SOX2* in day 6 and day 8 embryos. Expression of all the selected genes was detected by quantitative RT-PCR in both LPA-stimulated and control embryos (Figure 2A–F). The *YAP* expression levels were similar between LPA-stimulated and control embryos at days 6 and 8 (Figure 2A). The expression of *TAZ* and *TEAD4* was significantly higher in day 6 LPA stimulated blastocysts than in control embryos, but there was no difference between-treatment difference in day 6 morulae and day 8 blastocysts (Figure 2B,C). No significant differences in expression of *CDX2*, *OCT4* and *SOX2* were detected between LPA-stimulated and control embryos at days 6 and 8 (Figure 2D,E).

### 2.3. Effect of LPA on YAP and CDX2 Protein Expression

To investigate the localization of YAP, whole-mount immunofluorescence was performed on day 6 and day 8 bovine embryos. In agreement with the qRT-PCR results, YAP was detected in day 6 morulae and blastocysts, and in day 8 blastocysts (Figure 3). However, unlike in the mouse, we only detected nuclear YAP staining, even in cells spatially located in the inner part of the embryo (Figure 3). To assess the correlation between YAP and CDX2 localization and to evaluate YAP expression in ICM cells, embryos were simultaneously immunostained for CDX2. Interestingly, nuclear YAP was detected in both CDX2-positive and -negative cells in day 6 morulae and blastocysts (Figure 3), suggesting that nuclear YAP alone is not sufficient to drive CDX2 expression.

To examine whether LPA affects YAP expression, we determined the percentages of YAP expressing cells in day 6 and day 8 embryos. The total cell numbers were similar between embryos cultured with and without LPA for day 6 morulae, day 6 blastocysts and day 8 blastocysts (Figure 4A). A similar and high percentage of YAP positive cells was detected after culture with and without LPA in day 6 morulae (65.1% vs. 62.8% respectively) and day 6 blastocysts (77.6% vs. 78.1% respectively) (Figure 4B,C). By contrast, a significantly higher percentage of YAP positive cells was detected in day 8 blastocysts after LPA exposure (75.2%) compared to control medium (61.7%) (Figure 4B,C). The percentages of YAP positive cells were similar between day 6 and day 8 blastocysts after LPA exposure, but were significantly decreased from day 6 to day 8 in control conditions (Figure 4C), indicating that YAP degradation was inhibited by LPA exposure. To compare YAP expression in cells between embryos cultured with and without LPA, the average fluorescence intensity of a total of 5656 YAP positive cells from 98 embryos was analyzed. The fluorescence intensity, quantified as relative fluorescence units (RFU), was significantly higher when embryos were cultured with LPA, compared to control medium for day 6 morulae (83.5 RFU vs. 52.0 RFU), day 6 blastocysts (82.7 RFU vs. 72.2 RFU) and day 8 blastocysts (52.9 RFU vs. 44.5 RFU) (Figure 4D). 

Since absence of Hippo signaling can lead to CDX2 expression via YAP, we compared the percentages of CDX2 positive cells in day 6 and day 8 embryos cultured in the presence and absence of LPA. Contrary to expectations, the percentages of CDX2 positive cells were similar between embryos cultured with and without LPA for day 6 morulae (44.3% vs. 48.7%, respectively), day 6 blastocysts (63.9% vs. 65.7%, respectively) and day 8 blastocysts (68.2% vs. 68.7%, respectively) (Figure 4E,F). Moreover, the average CDX2 fluorescence intensity within nuclei were similar between embryos cultured with and without LPA for day 6 morulae (18.6 RFU vs. 15.8 RFU) and day 6 blastocysts (24.1 RFU vs. 23.8 RFU). However, the CDX2 intensity was significantly higher in day 8 blastocysts cultured in the presence of LPA (41.3 RFU), compared to control medium (24.9 RFU) (Figure 4G).

### 2.4. Colocalization of YAP and CDX2 in Bovine Embryos

We next focused on the relative localization of YAP and CDX2 in day 6 and day 8 embryos. A similar percentage of cells with colocalization of YAP and CDX2 was detected in embryos cultured, respectively, with or without LPA for day 6 morulae (43.3% vs. 43.7%), day 6 blastocysts (62.0% vs. 60.8%) and day 8 blastocysts (63.3% vs. 56.7%) (Figure 5A,B). In addition, the percentages of cells positive for YAP but negative for CDX2 expression was similar between embryos cultured with and without LPA for day 6 morulae (20.8% vs. 19.1%) and day 6 blastocysts (15.5% vs. 18.0%). However, for day 8 blastocysts, a significantly higher percentage of YAP positive but CDX2 negative cells was detected when embryos were cultured in the presence of LPA (11.5%), compared to control medium (4.9%) (Figure 5C,D). In addition, significantly lower percentages of CDX2 positive but YAP negative cells were present in embryos cultured with LPA, compared to control medium for day 6 morulae (1.1% vs. 5.0%), day 6 blastocysts (1.9% vs. 4.9%) and day 8 blastocysts (4.9% vs. 10.6%) (Figure 5E,F). Combined, the data suggest that CDX2 expression in the nucleus lags behind the nuclear localization of YAP.

## 3. Discussion

In mouse development, nuclear YAP is important for CDX2 expression and TE specification [13]. In human embryonic stem cells, however, YAP overexpression induces naïve pluripotency [18], indicating that the role of YAP in early cell lineage segregation may not be conserved among mammals. Since YAP/TAZ signaling can be enhanced by LPA [18,24,27], we studied the role of the Hippo/YAP pathway in embryonic development and lineage segregation by culturing bovine embryos in the presence of LPA. 

LPA is present in all mammalian cells and tissues and can induce a broad range of cellular effects, such as proliferation, survival, and migration [20,28,29]. Our results demonstrate that LPA stimulation accelerates bovine blastocyst formation on days 5 and day 6 of in vitro culture, without changing the total cell number at day 6 or day 8. This indicates that YAP signaling does not affect the rate of cell division during bovine embryo development. In mouse embryos, similarly, LPA did not alter cleavage rates but increased the percentage of 2-cell stage embryos that developed to blastocysts [30]. In porcine parthenogenetic embryos, LPA increased the cleavage and blastocyst rates and total cell numbers [31]; all of which suggests that the function of LPA in embryos varies between different mammalian species. 

In general, unphosphorylated YAP translocates to the nucleus, whereas phosphorylated YAP undergoes cytoplasmic inactivation [11,13,14]. Here we detected YAP only in nuclei in day 6 and day 8 bovine embryos, when using an anti-YAP antibody that recognizes both phosphorylated and unphosphorylated YAP; it is possible that YAP is rapidly removed from the cytoplasm after phosphorylation. 

LPA has been shown to increase YAP expression by inhibiting the Hippo pathway kinases LATS1/2 [18,24]. Indeed, by comparing the intensity of fluorescence, we showed that YAP expression levels were higher in day 6 and 8 embryos cultured in the presence of LPA than in embryos cultured without LPA. No difference in *YAP* gene expression was detected between embryos cultured with and without LPA, indicating that LPA does not affect *YAP* transcription. 

The transcription factor CDX2 is essential for TE specification and blastocyst formation [32]. Indeed, the percentage of CDX2 positive cells, and average CDX2 intensity within single nuclei, were higher in day 6 blastocysts than day 6 morulae. In the presence of LPA, the percentage of embryos that had developed into blastocysts at day 6 was almost doubled, indicating that more cells within those individual embryos reached the CDX2 expression threshold for TE differentiation in the presence of LPA. However, the intensity of fluorescence after CDX2 immunostaining was not different in LPA-exposed embryos, suggesting other mechanisms are involved in maintaining CDX2 expression at a certain level. Of the embryos that had reached the blastocyst stage at day 6, no difference in the percentage of CDX2-expressing cells was observed after LPA exposure, indicating that the TE/ICM ratio is not affected by the Hippo signaling pathway.

## 4. Materials and Methods

### 4.1. Chemicals

All chemicals were purchased from Sigma Aldrich (St. Louis, MO, USA) unless otherwise stated.

### 4.2. In Vitro Embryo Production and LPA Stimulation

Cumulus oocyte complexes (COCs) collection, oocyte recovery and in vitro fertilization were performed as described previously [33]. In short, COCs were aspirated from 2-8mm follicles from cattle ovaries, which were collected from a local slaughterhouse. the COCs were then cultured in maturation medium for 23 h at 38.5 °C, in an atmosphere of 5% CO_2_-in-air, after which oocytes were fertilized with sperm for 20–22 h and fertilization day was considered as day 0 of embryo development. After removal of the cumulus cells, zygotes were randomly allocated to one of two experimental groups: (i) cultured in synthetic oviductal fluid (SOF) [34]; and (ii) LPA, cultured in SOF supplemented with 10^−5^ M LPA for further development at 38.5 °C, in a humidified atmosphere containing 5% CO_2_ and 7% O_2_. On day 5, embryos were transferred to fresh SOF or SOF supplemented with 10^−5^ M LPA respectively, and cultured until day 8.

Developmental stage on day 5 and blastocyst percentage from days 5 to 8 were analyzed by 2 independent researchers (double-blinded).

### 4.3. RNA Isolation, cDNA Generation and Quantitative Reverse Transcription-PCR

Groups of 20–22 embryos were collected and stored in 100 µL RLT buffer (Qiagen, Venlo, The Netherlands) at −80 °C until RNA isolation. Total RNA isolation and cDNA generation was performed as described previously [34].

The quantitative reverse transcription PCR was performed as described previously [34] using specific primer sequences and annealing temperatures for amplification (Appendix A). Three independent biological cDNA samples were analyzed in duplicate. Expression of *RPL15*, *SDHA* and *YWHAZ* was used for normalization.

### 4.4. Immunofluorescence

Immunofluorescence was performed as described previously [34]. In short, after fixation, samples were incubated with mouse monoclonal antibody against CDX2 (Biogenex, CA, USA; CDX2-88; 1:200) and rabbit monoclonal antibody against YAP (Cell signaling technologies, Leiden, The Netherlands; #14074; 1:100) in dilution buffer (1× PBS, 1% BSA, 0.3% Triton X-100) overnight. Samples were incubated with the secondary antibodies: goat anti mouse Alexa488 (Invitrogen, Venlo, The Netherlands) and goat anti rabbit Alexa568 (Invitrogen, Venlo, The Netherlands) at 37 °C for 1 h after washings. Subsequently, nuclei were stained with DAPI for 20 min in the dark. After sample mounting, fluorescent images were obtained from a confocal laser microscope (SPE-II-DMI4000; Leica, Son, The Netherlands) with a Z-stack scanning and were further analyzed using IMARIS software (Bitplane, Zürich, Switzerland). Total cell number of individual embryo was determined by DAPI staining. Presumptive ICM cells and presumptive TE cells were determined by CDX2 negative and positive staining, respectively.

### 4.5. Statistical Analysis

Results are shown as means ± standard deviation in bar graphs. Excel and GraphPad Prism 8 (https://www.graphpad.com/scientific-software/prism/; accessed on 6 February 2019) was used to perform statistical analysis. Differences between two groups were analyzed by two-tailed unpaired Student’s *t*-tests. One-way ANOVA was performed to examine the differences between multiple groups and a post-hoc Tukey test was used to test significant differences for individual differences. Statistical significance was set at *p* < 0.05.

## 5. Conclusions

This study has demonstrated for the first time insightful and strongly evidence-based recognition of LPA-induced mechanisms underlying and identification of Hippo/YAP signaling pathway-related factors affecting not only onset and acceleration of blastocyst formation but also promotion of blastomere commitment and differentiation to ICM and TE cell lineages in bovine in vitro fertilization (IVF)-derived embryos. As a consequence, the LPA-triggered and Hippo/YAP pathway-mediated approaches could be widely applied to expedite the blastocyst formation by mammalian embryos that have been generated using other assisted reproductive technologies such as somatic cell nuclear transfer (SCNT) [35,36,37].

## Figures and Tables

**Figure 1 ijms-22-05915-f001:**
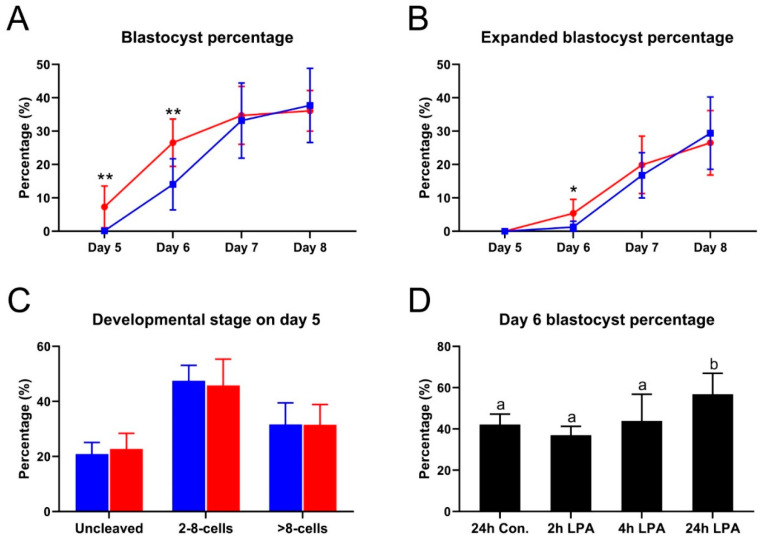
Effect of lysophosphatidic acid (LPA) on bovine embryo development. Blastocyst percentage (**A**) and expanded blastocyst percentage (**B**) after in vitro culture in the absence (blue) or presence (red) of 10^−5^ M LPA. Significant differences are individual time points are indicated by * (*p* < 0.05) and ** (*p* < 0.01). (**C**) Embryo development at day 5 after culture without (blue) and with (red) LPA. (**D**) Blastocyst percentage on day 6. Embryos with >8 cells were collected on day 5 and cultured without LPA (control) and with LPA for different time periods as indicated, and further cultured without LPA to complete the 24 h. Significant differences among columns are indicated by different letters above the bars. Data are depicted as mean ± standard deviation of more than three biological replicates. Con. = control, h = hours.

**Figure 2 ijms-22-05915-f002:**
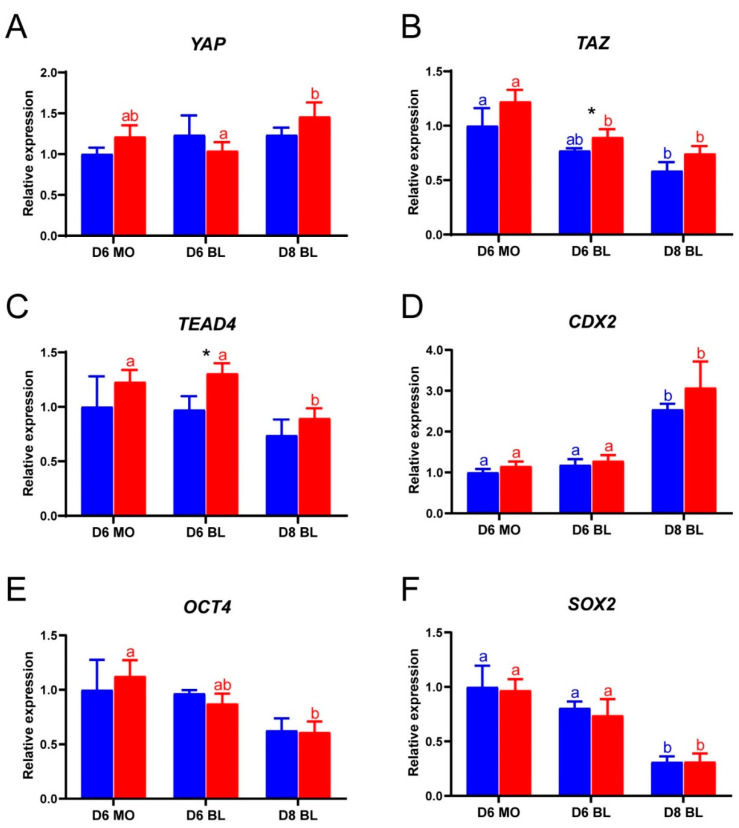
The relative expression of Hippo pathway and pluripotency genes in bovine embryos cultured in the absence of LPA (blue) and in the presence of 10^−5^ M LPA (red), as determined by quantitative RT-PCR. (**A**) *YAP*, (**B**) *TAZ*, (**C**) *TEAD4*, (**D**) *CDX2*, (**E**) *OCT4*, (**F**) *SOX2*. Relative expression from control embryos was set at 1; * (*p* < 0.05) indicates a significant difference between embryos cultured with and without LPA. Significant differences between developmental stages are presented by different letters with the same color (*p* < 0.05). Error bars indicate standard deviations of three biological replicates. D = day, MO = morula, BL = blastocyst.

**Figure 3 ijms-22-05915-f003:**
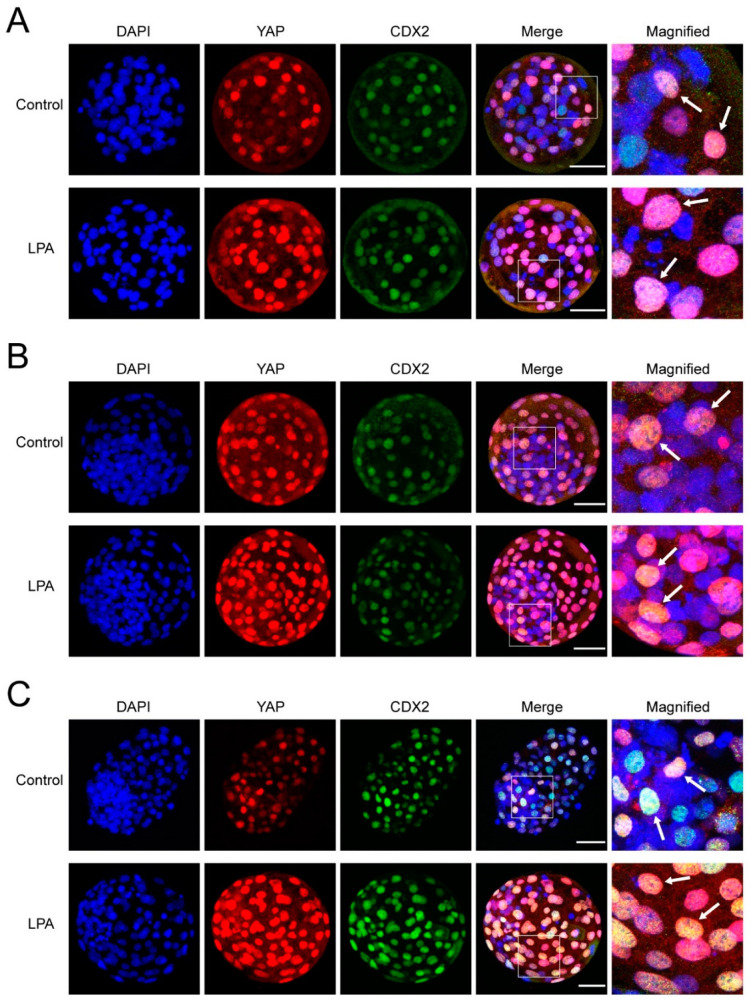
YAP and CDX2 immunofluorescence in bovine embryos cultured in the absence (control) or presence of 10^−5^ M LPA. Day 6 morula (**A**), day 6 blastocyst (**B**) and day 8 blastocyst (**C**). The areas in the white boxes are presented at higher magnification in the right column. YAP and CDX2 colocalization is indicated (arrows). Scale bar = 50 μm.

**Figure 4 ijms-22-05915-f004:**
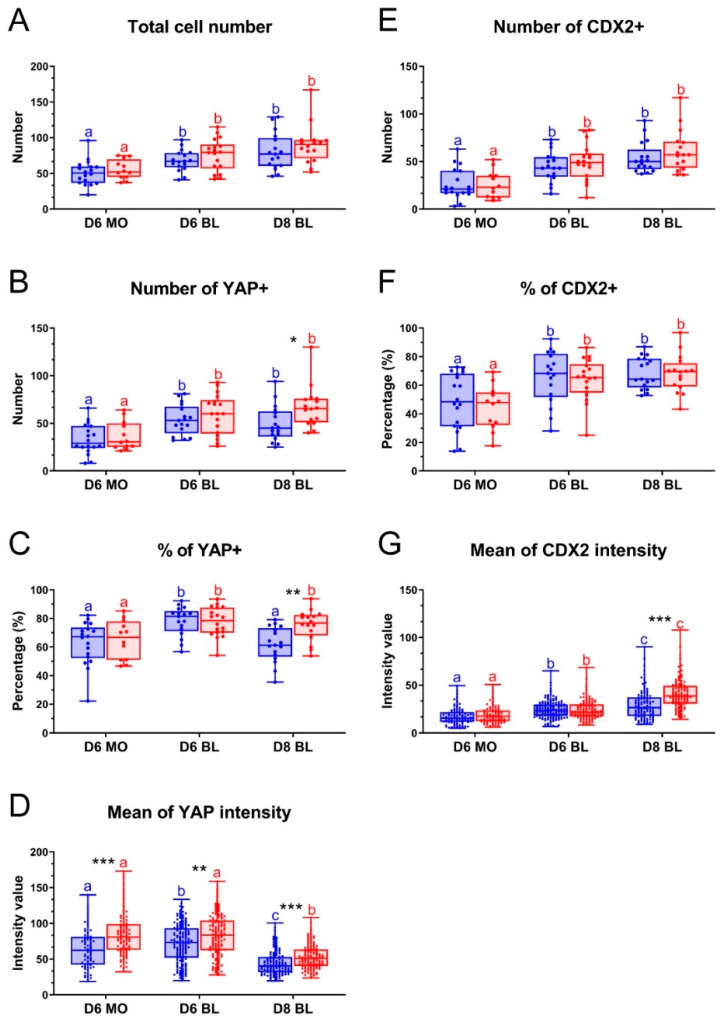
Effect of LPA on YAP and CDX2 expression in bovine embryos. Embryos were cultured in the absence of LPA (blue) or in the presence of 10^−5^ M LPA (red). Total cell number (**A**), number of YAP positive cells (**B**), percentage of YAP positives out of total cell population (**C**), Mean YAP intensity (RFU) in single nuclei (**D**), number of CDX2 positive cells (**E**), percentage of CDX2 positives of total cell population (**F**), Mean CDX2 intensity (RFU) in single nuclei (**G**). * (*p* < 0.05), ** (*p* < 0.01) and *** (*p* < 0.005) indicate significant differences between embryos cultured with and without LPA. Significant differences among embryos cultured under the same conditions are indicated by different letters with the same color (*p* < 0.05). Error bars indicate standard deviations of embryos or single cells, collected from three biological replicates. D = day, MO = morula, BL = blastocyst.

**Figure 5 ijms-22-05915-f005:**
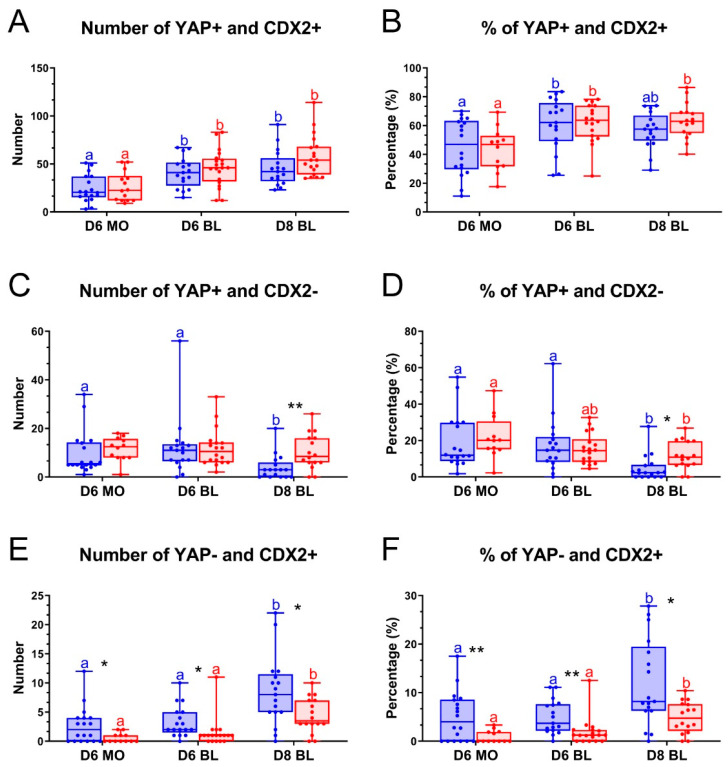
Effect of LPA on YAP and CDX2 localization in bovine embryos. Embryos were cultured in the absence of LPA (blue) or in the presence of 10^−5^ M LPA (red). Number of cells with colocalized YAP and CDX2 (**A**), percentage of YAP and CDX2 colocalization out of total cells (**B**), number of YAP positive (YAP+) but CDX2 negative (CDX2-) cells (**C**), percentage of YAP positive but CDX2 negative cells (**D**), number of YAP negative (YAP-) but CDX2 positive (CDX2+) cells (**E**), percentage of YAP negative but CDX2 positive cells (**F**). * (*p* < 0.05) and ** (*p* < 0.01) indicate significant differences between embryos cultured with and without LPA. Significant differences among embryos cultured under the same conditions are indicated by different letters with the same color (*p* < 0.05). Error bars indicate standard deviations for embryos, collected from three biological replicates. D = day, MO = morula, BL = blastocyst.

## Data Availability

Data is contained within the article.

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
