# Peer review of "Lysophosphatidic Acid Accelerates Bovine In Vitro-Produced Blastocyst Formation through the Hippo/YAP Pathway"

_ijms, 2021, doi:10.3390/ijms22115915_

Round 1

Reviewer 1 Report

The authors of "LPA accelerates blastocyst formation......." have submitted an interesting paper deciphering the role of Hippo/Yap pathway in the blastocyst formation of bovine mammalian embryos. Although all mammalian embryos share common molecular pathways during preimplantation development there exist minor or major differences in their contribution to the acceleration or formation of blastocysts.

Hippo is already known to play a role in follicles and therefore is expected to contribute in early embryos but this to my opinion is worth mentioning in the introduction or discussion. In addition the blastocyst cell counting method to determine the TE/ICM ratio is not clearly mentioned in the methods.

Reviewer 2 Report

11th May, 2021

Review of the Manuscript ID ijms-1217136, by B. Yu et al., entitled: “Lysophosphatidic acid accelerates blastocyst formation through the Hippo/YAP pathway” that is intended to be published as the Article in International Journal of Molecular Sciences

Taking into account research highlight, contribution of the Authors to the progress in the research area, thorough manner of data presentation, perfectly writing in English, abundance of Results and Figures (diligent graphic and photographic documentation), the quality of this paper deserves praise and merits my support. The Authors have received the very high scores from me for the originality, significance of the work and the scientific value of their paper. In my opinion, the current paper provides insightful interpretation of topical and coming trends in explanation of molecular mechanisms underlying and identification of molecular factors that determine not only initiating and expediting blastocyst formation but also committing and differentiating blastomeres to inner cell mass (embryoblast) and trophoblast cell lineages in bovine in vitro fertilization (IVF)-derived embryos. For all these above-mentioned reasons, this original article deserves to be recognized and supported by the Reviewer. Therefore, I strongly recommend the Editorial Board to allow for publication of this excellent paper in International Journal of Molecular Sciences, after the minor revision of the manuscript will have been completed by the Authors and provided that the Authors are ready to consider all the Reviewer comments/remarks detailed below:

1) In my opinion, the present version of article title: “Lysophosphatidic acid accelerates blastocyst formation through the Hippo/YAP pathway” should have been modified to the one of the following options suggested by the Reviewer:

  1. a) “Lysophosphatidic acid accelerates via Hippo/YAP pathway the blastocyst formation by bovine in vitro-fertilized embryos

or

  1. b) “Lysophosphatidic acid accelerates bovine in vitro-produced blastocyst formation through the Hippo/YAP pathway

2) Moreover, there is a lack of the Conclusions section in the article. That is why, this subsection should have been added at the end of manuscript text as has been indicated below:

  1. Conclusions

This study has demonstrated for the first time insightful and strongly evidence-based recognition of LPA-induced mechanisms underlying and identification of Hippo/YAP signaling pathway-related factors affecting not only onset and acceleration of blastocyst formation but also promotion of blastomere commitment and differentiation to ICM and TE cell lineages in bovine in vitro fertilization (IVF)-derived embryos. As a consequence, the LPA-triggered and Hippo/YAP pathway-mediated approaches could be widely applied to expedite the blastocyst formation by mammalian embryos that have been generated using other assisted reproductive technologies such as somatic cell nuclear transfer (SCNT) [31–33].

3) Furthermore, the following 3 References have to be added and cited in the text of manuscript (according to the re-editions/changes required by Reviewer in the above-listed comment 2):

  1. Zhang, Y.; Gao, E.; Guan, H.; Wang, Q.; Zhang, S.; Liu, K.; Yan, F.; Tian, H.; Shan, D.; Xu, H.; Hou, J. Vitamin C treatment of embryos, but not donor cells, improves the cloned embryonic development in sheep. Reprod. Domest. Anim. 2020, 55, 255265, doi:10.1111/rda.13606.
  2. Samiec, M.; Skrzyszowska, M.; Opiela, J. Creation of cloned pig embryos using contact-inhibited or serum-starved fibroblast cells analysed intra vitam for apoptosis occurrence. Ann. Anim. Sci. 2013, 13, 275–293, doi:10.2478/aoas-2013-0009.
  3. Olivera, R.; Moro, L.N.; Jordan, R.; Luzzani, C.; Miriuka, S.; Radrizzani, M.; Donadeu, F.X.; Vichera, G. In vitro and in vivo development of horse cloned embryos generated with iPSCs, mesenchymal stromal cells and fetal or adult fibroblasts as nuclear donors. PLoS One 2016, 11, e0164049, doi: 10.1371/journal.pone.0164049.

4) Additionally, there is a lack of the Abbreviations section in the article. That is why, this subsection should have been added by the Authors to the manuscript (before the References subsection).

General Comment of the Reviewer:

Before the manuscript will have been accepted for publication in International Journal of Molecular Sciences, it requires the minor revision (according to all the remarks and suggestions of the Reviewer) and re-review to confirm the correctness of changes that will have been made by the Authors in the re-edited and resubmitted version of their paper. 
